# Spinal Cord Injury: Pathophysiology, Multimolecular Interactions, and Underlying Recovery Mechanisms

**DOI:** 10.3390/ijms21207533

**Published:** 2020-10-13

**Authors:** Anam Anjum, Muhammad Da’in Yazid, Muhammad Fauzi Daud, Jalilah Idris, Angela Min Hwei Ng, Amaramalar Selvi Naicker, Ohnmar Htwe@ Rashidah Ismail, Ramesh Kumar Athi Kumar, Yogeswaran Lokanathan

**Affiliations:** 1Centre for Tissue Engineering and Regenerative Medicine, Faculty of Medicine, Universiti Kebangsaan Malaysia, Jalan Yaccob Latiff, Cheras, Kuala Lumpur 56000, Malaysia; anamanjum40@gmail.com (A.A.); dain@ukm.edu.my (M.D.Y.); angela@ppukm.ukm.edu.my (A.M.H.N.); 2Institute of Medical Science Technology, Universiti Kuala Lumpur Malaysia, Kajang 43000, Malaysia; mfauzid@unikl.edu.my (M.F.D.); jalilahidris@unikl.edu.my (J.I.); 3Department of Orthopaedics & Traumatology, Faculty of Medicine, Universiti Kebangsaan Malaysia, Kuala Lumpur 56000, Malaysia; amara@ppukm.ukm.edu.my (A.S.N.); ohnmar@ppukm.ukm.edu.my (O.H.R.I.); 4Department of Surgery, Universiti Kebangsaan Malaysia Medical Centre, Jalan Yaacob Latiff, Bandar Tun Razak, Kuala Lumpur 56000, Malaysia; rameshkumar@ppukm.ukm.edu.my

**Keywords:** spinal cord injury, primary injury, secondary injury, neurodegeneration, neuroprotection, neuro-regeneration

## Abstract

Spinal cord injury (SCI) is a destructive neurological and pathological state that causes major motor, sensory and autonomic dysfunctions. Its pathophysiology comprises acute and chronic phases and incorporates a cascade of destructive events such as ischemia, oxidative stress, inflammatory events, apoptotic pathways and locomotor dysfunctions. Many therapeutic strategies have been proposed to overcome neurodegenerative events and reduce secondary neuronal damage. Efforts have also been devoted in developing neuroprotective and neuro-regenerative therapies that promote neuronal recovery and outcome. Although varying degrees of success have been achieved, curative accomplishment is still elusive probably due to the complex healing and protective mechanisms involved. Thus, current understanding in this area must be assessed to formulate appropriate treatment modalities to improve SCI recovery. This review aims to promote the understanding of SCI pathophysiology, interrelated or interlinked multimolecular interactions and various methods of neuronal recovery i.e., neuroprotective, immunomodulatory and neuro-regenerative pathways and relevant approaches.

## 1. Introduction

Spinal cord injury (SCI) is a devastating neurological state producing physical dependency, morbidity, psychological stress and financial burden. For the last 30 years, its global prevalence has increased from 236 to 1298 cases per million populations. The estimated global rate of SCI falls between 250,000 and 500,000 individuals every year [1]. The total lifetime costs for each patient with SCI exceed 3 million dollars, and the calculated annual economic burden is almost 2.67 billion dollars in Canada [2]. Available treatments are limited and only provide supportive relief to patients with lifetime disability [1]. Heterogeneous factors such as complex characteristics, abundant inconsistencies and complex pathophysiologic consequences post-SCI are the major reasons for poor understanding and failure of SCI treatment. Hip joint subluxation caused by SCI is challenging to overcome and causes lower leg paralysis [3]. SCI is also associated with autonomic dysreflexia (AD) occurring in 48%–60% of cases at above thoracic 6th vertebral level (T6) and involving a sudden onset of excessively high blood pressure [4]. Understanding pathophysiology, phases and various wound recovery mechanisms associated with SCI is essential for the development of appropriate recovery treatments [5]. Normal spinal cord physiology involves interactions among many cell types such as astrocytes, neurons, microglia and oligodendrocytes. After a spinal injury, these multicellular interactions are interrupted and disorganised, leading to an impaired spinal recovery [5]. Various animal studies showed that the administration of current SCI treatments such as drugs, neuronal implants and stem cells induced the following improvements: (i) decrease neuro-inflammation, (ii) promote axonal growth, (iii) enhance myelination and (iv) reduce cavity size [2]. However, the current treatment strategies can aid for only a short duration and fail to completely overcome the detrimental effects of SCI. Therefore, knowledge on fundamental SCI pathophysiology and event sequences during and post-injury is beneficial in designing a suitable intervention for SCI [5]. Despite numerous studies and availability of various regenerative treatment strategies, post-SCI recovery remains controversial, and scientists are still exploring methods that could prevent or reverse the devastating outcomes of SCI [2]. This review highlights recent findings and critical knowledge gaps about fundamental pathophysiology following SCI, multicellular and multimolecular interactions, phases and underlying recovery mechanisms of SCI, especially those targeting neuroprotection, immuno-modulatory and neuro-regenerative pathways. Strategies to re-establish the lost connectivity between spinal cord cells and their interactions are also explored.

### 1.1. SCI Phases

#### 1.1.1. Primary Injury

Acute SCI commonly occurs due to sudden trauma to the spine and results in fractures and vertebrae dislocation. The initial stage immediately after the injury is known as primary injury [2,4] (Figure 1a) with features of bone fragments and spinal ligament tearing. SCI is accomplished in two phases: the first phase includes the destruction of neural parenchyma, disruption of axonal network, haemorrhage and disruption of glial membrane (Figure 1a). The main determinants for SCI severity are the extent of initial destruction and duration of spinal cord compression. A cascade of events associated with secondary injury is activated by the onset of biochemical, mechanical and physiological changes within neural tissues [6]. Although clinical manifestation suggests complete functional loss, few segments remain connected by some axons during primary SCI phase, thus reflecting incomplete and partial injury state [6,7].

#### 1.1.2. Secondary Injury

The primary injury triggers secondary injury which produces further chemical and mechanical damage to spinal tissues, leads to neuronal excitotoxicity because of high calcium accumulation within cells and increases reactive oxygen concentrations and glutamate levels. These incidences damage underlying nucleic acid, proteins and phospholipids and result in neurological dysfunction [7]. The secondary injury phase reflects multi-featured pathological processes following the primary injury phase and lasts for several weeks (Figure 1b). Clinical manifestation of secondary injury includes increased cell permeability, apoptotic signalling, ischemia, vascular damage, oedema, excitotoxicity, ionic deregulation, inflammation, lipid peroxidation, free radical formation, demyelination, Wallerian degeneration, fibroglial scar and cyst formation as shown in Figure 2 [7]. Disruption of blood vessels causes haemorrhage in spinal tissues, followed by invasion of monocytes, neutrophils, T and B lymphocytic cells and macrophages to spinal tissues. This phenomenon is also associated with the release of inflammatory cytokines such as interleukin (IL)-1a, IL-1b, IL-6 and tumour necrosis factor (TNF)-α after 6–12 h post-injury. The penetration of immune cells and inflammatory cytokines promotes the inflammation of neurons [8].

The secondary injury is categorised into three phases: acute, sub-acute and chronic injury (Figure 1b). Following the primary injury phase, the initiation of acute secondary injury phase begins is manifested through clinical features such as vascular damage, ionic imbalance, excitotoxicity, free radical production, increased calcium influx, lipid peroxidation, inflammation, oedema and necrosis [9]. If the acute secondary injury phase persists, then the sub-acute secondary injury phase begins and is manifested by features such as neuronal apoptosis, axonal demyelination, Wallerian degeneration, axonal remodelling and glial scar formation [9] as shown in Figure 2. Sub-acute secondary injury leads to the chronic secondary injury phase of SCI as characterised by the formation of cystic cavity, axonal dieback, and maturation of glial scar [10].

### 1.2. Pathophysiology of SCI

SCI pathophysiology comprises interrelated events, each serving as the facilitator for the other. In some instances, multiple events occur simultaneously and cause complicated attributes, thus rendering this illness difficult to treat. SCI can be represented as a cascade of different interrelated events (Figure 2).

The most vulnerable clinical manifestation immediately after injury is the interruption of spinal cord vascular supply and hypotension/hypo-perfusion, producing hypovolemia, neurogenic shock and bradycardia. These signs occur because of extensive bleeding and neurogenic shock leading to spinal cord ischemia. The rupture of small blood vessels and capillaries promotes the extravasation of leukocytes and red blood cells (RBCs). These extravasations of immune cells at the injury site exert pressure on the injured spinal tissues and further disrupt the blood flow, thus producing vasospasm [9]. This state continues up to 24 h. Occurrence of vascular ischemia, hypovolemia and hyper-perfusion eventually leads to cell death and tissue destruction [9,10].

Spinal cord ischemia causes cytotoxic, ionic and vasogenic oedemas. In normal physiology, the influx of Na^+^ occurs due to the passive influx of Cl− through chloride channels. Consequently, water molecules influx through aquaporin water channels. During a pathophysiological state, the balance between solute and water influx at the intracellular compartment is disturbed, thereby causing cell swelling and loss of cytoskeletal integrity and promoting cell death [11]. Ionic oedema occurs due to the increased permeability of the blood–spinal cord barrier that increases trans-endothelial ion transport and causes the loss of ions and water from the interstitial space [12]. Endothelial injury and inflammation subsequently increase the pore size and thus allow large plasma-derived molecules to pass through the cell membrane, resulting in vasogenic oedema [12]. This acute secondary injury phase continues from 2 h to 48 h. Continuous haemorrhage, oedema and inflammatory stage lead to substantial necrosis indicated by the increased concentration of specific inflammatory and the presence of structural biomarkers, e.g., glial fibrillary acidic protein (GFAP) or IL-6 in cerebrospinal fluid (CSF) [6]. These processes provoke free radical formation, glutamate-mediated excitotoxicity and neurotoxicity [12] (Figure 1c).

Glutamate is an excitatory neurotransmitter that is released in the central nervous system (CNS) and interacts with N-methyl-D-aspartate (NMDA), α-amino-3-hydroxy-5-methyl-4-isoxazolepropionic acid (AMPA) and kainate ionotropic and metabotropic receptors [12] (Figure 2). The activation of glutamate receptors during SCI greatly increases glutamate concentrations and produces persistent excitotoxicity and cell death [12]. Abnormal increases in glutamate excitation are caused by diverse events, such as mechanical stress, formation of apoptotic and necrotic cells, failure of Na^+^/K^+^ ATPase in the axonal membrane, lipid peroxidation and formation of 4-hydroxynonenal [5]. Hyper-activation of NMDA and AMPA receptors increases the influx of Ca^2+^ and Na^+^ ions which further promotes apoptosis and necrotic cell death [12].

High levels of glutamate in necrotic cells alter the ionic flux by increasing intracellular Na^+^ and Ca^2+^ concentrations and decreasing intracellular K^+^ concentrations. An increase in Ca^2+^ concentration inhibits mitochondrial respiration and energy depletion and consequently disturbs ionic homeostasis. Alteration in the function of Na^+^/K^+^ ATPase elevates axonal membrane depolarisation and leads to excessive Na^+^ influx within axon membranes. This ionic dysregulation causes cell cytotoxic oedema, axonal acidosis, increased Ca^2+^ membrane permeability, activation of phospholipases, increased reactive oxygen species (ROS) generation and mitochondrial dysfunction [11,12] (Figure 1c).

Mitochondria are an integral component for cellular metabolism because they generate ATP (Adenosine triphosphate) molecules through phosphorylation. These organelles have four components, i.e., an outer mitochondrial membrane (OMM), inner mitochondrial membrane (IMM), intermembrane space (IMS) and inner matrix. OMM regulates the passage of molecules via voltage-dependent anion channels (VDAC) and maintains a potential of 5 kDa, and IMM controls the exchange of oxygen, water and carbon dioxide [13]. The electron transport chain (ETC) regulates the proton gradient within mitochondria and comprises NADH dehydrogenase (complex 1) and ATP synthases (complex V). Complex 1 oxidises NADH and produces energy. CNS cells contain a large number of complex 1 and generate ROS. Coenzyme Q and cytochrome regulate electron transport in ETC. This transportation and the control of electrons reduce the production of ROS. Complex V generates ATP, acts as a proton channel, converts ADP to ATP and utilises ATP to pump back protons to intermembrane space, hence utilising energy in place of producing ATP [13]. Mitochondria also work as energy reservoirs, regulate cytosolic Ca^2+^ levels and serve as a vital role in calcium-dependent neuronal death [8]. In SCI, elevated cytosolic Ca^2+^ levels activate the complex 1, increase ATP generation and promote ROS production. Ca^2+^ passes the mitochondria through the mitochondrial calcium uniporter [9]. The accumulation of high cytosolic Ca^2+^ leads to membrane permeabilisation and increases mitochondrial permeability transition pores (mPTPs) [13]. The opening of mPTPs disturbs the proton gradient, inactivates ATP production increases the influx of water and other components within a mitochondrial matrix, and results in cell swelling and finally death [13,14] (Figure 2). Calcium overload also promotes protein kinases and phospholipases which cause calpain-associated protein degradation and oxidative damage [9,15]. Most of the energy required by brain is provided by mitochondria, and sufficient energy is required for neuronal survival; therefore, mitochondrial dysfunction could result in neuronal death [16].

High ROS and reactive nitrogen species (RNS) generation induces various deleterious effects, including lipid peroxidation on different body organs. Lipid peroxidation transpires in three steps: (i) ROS reacts with the membrane’s polyunsaturated fatty acid component and snatches an electron from it. This electron binds to lipid molecules and generates reactive lipid species (ii) which quenches other radicals, generates additional reactive species and (iii) finally produce other reactive species including 4-hydroxynonenal (HNE) and 2-propenal [17]. Neuroinflammation is a key process associated with SCI and involved numerous cell types such as neutrophils, microglia, macrophages, astrocytes, dendritic cells (DCs) and B-and T-lymphocytes and molecular components such as cytokines and prostanoids [17,18]. The complex inflammatory responses following SCI produce neurotoxic or neuroprotective effects depending on the duration and time of responses. Early inflammatory cells and mediators such as macrophages may also have beneficial functions by assisting in inflammation, repair and recovery [18]. Apoptosis and necrosis are vital cell death processes in SCI. In 2012, the Nomenclature Committee on Cell Death lists 12 different types of cell death mechanisms such as necroptosis, pyroptosis, autophagy and netosis [19,20]. During apoptosis, the cell shrinks, followed by phagocytosis [21,22] (Figure 2). Another major process that mediates cell death is autophagy [23] which works as a recycling agent and detoxifies unwanted proteins and organelles by promoting autophagosomal and lysosomal pathways. During SCI, the abnormal activation of autophagosomes and lysosomes triggers rapid cell death [24]. Few other mechanisms of cell death such as programmed cell death called necroptis [25], regulated cell death calledparthanatos [26] and caspase-independent cell death pathways often involving apoptosis-inducing factor (AIF) [21] are not clearly understood and need further investigations. Necroptosis is a programmed necrotic cell death playing a vital role in neuronal cell death [25]. The detailed explanation of ROS and RNS generation, apoptotic pathways and neuroinflammation is presented in the following section.

Acute axonal degeneration (AAD, Figure 3) is another important clinical manifestation of early acute SCI phase. This process induces other effectors such as cysteine protease calpain and Wallerian degeneration which further promote axonal degeneration [27]. AAD is initiated by a high Ca^2+^ influx into axons. A high Ca^2+^ deposition increases AAD risk in axons [27]. This phenomenon occurs in two phases, the earlier phase occurs within 15 min post-injury, and the later phase called Wallerian degeneration occurs after a few hours (24–48 h) [28]. The Wallerian degeneration is manifested by the formation of retraction bulbs, a microtubule network that inhibits axonal regeneration [28]. The anterograde degenerative mechanism is termed as Wallerian degeneration; however, retrograde degeneration of axons is termed as axonal dieback [6].

Demyelination occurs when myelin, the protective coating of nerve cells, is damaged. This process slows down the messages sent along axons and deteriorates axon and oligodendrocytes [29]. Oligodendrocytes are myelinating cells that promote the proliferation and myelination of axons [30] and are sensitive to glutamate excitotoxicity that occurs due to the hyperactivation of AMPA, kainate and NMDA receptors [11]. During SCI, oligodendrocytes undergo necrosis and apoptosis. A high glutamate level increases Ca^2+^ influx that provokes cell death [11]. Damage to oligodendrocytes is also induced by ROS and RNS production, glutathione reduction, and increase in iron load and peroxisome hyperactivation [31]. ROS production by neutrophils and microglia triggers the release of pro-inflammatory cytokines such as TNFα, IL-2 and interferon (IFN) γ and proteases and further facilitates oligodendrocyte apoptosis [31]. The formation of pro-inflammatory cytokines such as TNFα plays a vital role in the inflammation and apoptosis of oligodendrocytes [31]. The apoptosis of oligodendrocytes causes the demyelination of axon and results in the loss of axonal function and stability because single oligodendrocytes myelinate several other axons [31,32]. The demyelination of oligodendrocytes also induces the expression of Fas-receptors that release caspases 3 and 8 which mediate the apoptosis [22,23].

Glial scar formation (gliosis) (Figure 2) is a reactive cellular mechanism that is facilitated by astrocytes and occurs during the chronic secondary phase of SCI. The scarring of astrocytes (astrogliosis) is the body’s natural process that shields and starts the healing post-SCI [33]. Astrocytes are an important component of the nervous system. The astrocytes are sensitive towards changes such as alteration in gene expression, hypertrophy, and excitations [34]. The other major constituents of the scar tissue are pericytes and the connective tissues. In normal physiology, the number of astrocytes is 10 times higher in spinal cord parenchyma that that of pericytes. However, 2 weeks after post-injury, the pericytes are twice the number of astrocytes [34]. Pericytes secrete specific markers that promote fibroblast to express ECM such as fibronectin which serves as the main component of scar connective tissues [35].

The continuous enlargement of lesion site and formation of the cyst is the hallmark feature of SCI. The formation of cyst reveals ongoing apoptotic responses while astrocytes undergo necroptosis cell death through TLR4/MyD88 signalling [36]. Cyst formation leads to syringomyelia in approximately one-third of patients with SCI. Syringomyelia is a condition in which a cyst (syrinx) or cavity develops within the spinal cord, progresses over time and damages the spinal cord. The destruction may result in sensation loss, paralysis, weakness and stiffness in the back, shoulders and extremities [37]. The complications related to syringomyelia are often observed in SCI, but the pathophysiology of syrinx formation is poorly understood [37].

### 1.3. Multicellular and Multi-Molecular Interactions

Multicellular interactions play an important role in developing effective neuroprotective and neurodegenerative strategies to overcome detrimental outcomes following SCI. The pathophysiology of SCI and the multicellular interactions between neuronal cells, neuroglia cells and non-neuronal cells must be understood to outline effective protective and regenerative strategies for SCI. Anatomically SCI is partitioned into complete and incomplete injury. A complete injury is referred to as a condition in which SCI is severe and the complete loss of function at and below the injury site. This requires the restoration of neural connectivity all along the lesion core and can occur due to a single large lesion or multiple small lesions, which make it difficult to build connections. For incomplete injury, the activity of the spinal cord is compromised but the brain’s ability to send signals and messages below the injury site is not completely lost [3]. This condition manifests as a small lesion that consists of structures controlling several activities such as neural protection and restoration of functions [38]. In SCI, the synaptic and circuit reorganisation that occur post-SCI produce adaptive and maladaptive functional changes. Spontaneous synapses also transpire and sometimes act with circuit reorganisation to cause muscle spasticity, autonomic dysreflexia and neuropathic pain [38]. Hence, understanding cellular and molecular mechanisms and interactions is essential to devise strategies that restore circuit reorganisation.

Axons are the main element of the neuron that is considered during treatments and recoveries following SCI. Effective treatment strategies depend on a thorough understanding of axon growth and cellular responses and how these responses are modulated by specific molecular and cellular mechanisms in each stage of pathophysiologic response. The degree of response can be differentiated by phases such as: (i) axon degeneration and retraction, (ii) axon regeneration at fibrotic scars, (iii) axon regeneration at viable neural tissues, (iv) axon sprouting and (v) local synaptic plasticity [39].

Cells involved in damage and repair process during SCI can be divided into two main groups for ease of consideration such as (i) neural and non-neural intrinsic cells and (ii) blood-borne non-neural cells [40]. Neural intrinsic cells are neurons, oligodendrocytes, astrocytes, and neuron glial antigen 2 oligodendrocyte progenitor cells (NG2-OPCs). The intrinsic non-neural cells such as microglia stimulate phagocytic responses, perivascular fibroblasts, pericytes, and endothelial progenitors [40]. The endothelial progenitor cells produce laminin (a growth regulator) that helps in the migration of cells and axons. The blood-borne cells include leukocytes, platelets, fibrocytes and mesenchymal stem cells. The post SCI these blood-borne cells migrates to the injury site, embeds in the extracellular matrix (ECM), and contributes to the repair and regeneration of injured tissues. Various ECM components such as laminin, collagens, and glycoproteins such as chondroitin or heparan sulphate proteoglycans (CSPGs or HSPGs), also enhance tissue repair and axonal regeneration [41].

Tissue regeneration is divided into three overlapping distinct phases, i.e., (i) cell death and inflammation, (ii) cell proliferation, and tissue replacement and (iii) tissue remodelling. Cell death and inflammation: the first event after SCI is haemostasis to stop blood loss through coagulation cascade, platelet aggregation, and clot formation. Inflammatory and immune cells migrate to the injured site and perform phagocytosis to remove cell debris. The platelet aggregates also provide support to the migrating neutrophils, macrophages and leukocytes at the SCI site. The endogenous mesenchymal cells enter the lesion core and facilitate tissue repair responses [42]. Microglia and NG2-OPC also migrate towards the lesion core to participate in regeneration. The astrocytes remain outside the lesion core. Different processes simultaneously occur in overlapping sequences during the first few days after injury [42]. Cell proliferation and tissue replacement: these responses take place after 2–10 days following the injury to repair and regenerate tissues. Cells that are involved in this phase include endothelial progenitor cells, fibroblast, inflammatory cells, glial and neural progenitor cells, and scar-forming astrocytes. In this phase, many proliferative mechanisms take place such as (i) proliferation of endothelial cells, (ii) fibroblast linkage cells, and inflammatory cells causing astrocytes scar formation. Several proteins diffuse in neural parenchyma such as serum proteins (e.g., thrombin and albumin), immunoglobulins, and pathogen-associated molecules [43]. During the proliferative phase, the lesion can be represented by two tissue compartments (i) central non-neural lesion core and (ii) astrocytes scars surrounding the lesion. This proliferative phase is notified by the location of the astrocyte scar border, which can be differentiated as a separate non-functioning persisting area surrounding the functioning neural tissues [44]. The location of astrocytes scar borders surrounding the lesion and the associated multicellular and molecular interaction is important in devising therapeutic strategies to reduce the lesion size [45]. During the proliferative phase, different neural progenitor cells migrate to an injury site, and evidence has shown that the viable cells surrounding astrocytes contribute actively in tissue remodelling and immune regulation [44]. Tissue remodelling: this phase starts after the first-week of post-injury and can be distinguished by the formation of new blood vessels and the presence of functional astrocytes and pericytes. A mature and compact astrocyte scar after 2–3 weeks will be surrounded by a non-neural lesion core to limit the lesion tissues [46]. Hence, the core can be differentiated into three compartments: (i) central non-neural core (fibrotic scar), (ii) astroglial scar and (iii) perilesion perimeters (Figure 4). This compact astrocyte scar serves as a protective coat that limits inflammatory cell migration from non-neural lesion core to the surrounding viable neural cells [33,46].

#### 1.3.1. Fibrotic Scar

Following SCI various cellular events such as excitotoxicity, ROS generation, metabolic derangement damage, hypoxia and ischemia occur. This cascade of events produces cellular debris, which may be toxic. Multiple mechanisms are initiated to clear this cellular debris to protect the healthy cells from damage [47]. Microglia and astrocytes act as early responders that perform required phagocytic activity and activate growth factors, cytokines and blood-borne inflammatory cells to eliminate the toxic debris (Figure 4). The immune system is responsible for protecting the healthy cells from toxin damage, but these responses should be balanced. The balanced inflammatory response involves pro-inflammatory (M1) and anti-inflammatory (M2) responses that are essential since too extreme response such as slower response in inflammation results in cytotoxins accumulation and higher inflammatory response results in cellular damage [48]. M1 responses promotes antigen for T cells and activate, phagocytosis, innate, and adaptive immune responses, while M2 responses reduces NF-κB pathway activity and in return reduces inflammation [48]. Lesion core is composed of fibroblast-derived stromal cells, meningeal fibroblasts, and pericytes. The composition of mature lesions includes non-neural cells, fibrocytes, blood vessels, macrophages, neutrophils, lymphocytes and leukocytes embedded in ECM [48,49].

#### 1.3.2. Astroglial Scar

After 7–10 days of SCI, astrocytes proliferate and assemble along the margin of extensively damaged tissue. Then, these freshly proliferated astrocytes migrate and organise as a scar border margining the swollen non-neural lesion core tissue (fibrotic scars). This covering of scar by astrocytes is completed in 2–3 weeks post SCI [47]. The main role of these astrocyte covering is to keep inflammatory cells within damaged tissue area and protect surrounding viable neural tissue from destructive inflammatory phagocytosis [46]. The astrocytes scar is only several cell layers thick. The reactive oligodendrocytes interact with astrocytes cells on scar borders and produce the oligodendrocytes progenitor cells (OPC) that releases neuron glial antigen 2 (NG2), also referred as the chondroitin sulphate proteoglycan 4 (Figure 4). The NG2 which expresses OPC cells are termed as NG2-OPC [47].

#### 1.3.3. Perilesion Perimeters

The viable cells surrounding astrocytes scar layer that have normal physiology. They are composed of multicellular components such as reactive glia, astrocytes, microglia, NG2-OPCs, and oligodendrocytes [48] (Figure 4). The astrocytes can be classified as (i) the hypertrophic reactive astrocytes and (ii) the mature astrocytes. The astrocytes follow normal interaction with active neurons while hypertrophic reactive astrocytes are phenotypically and functionally distinct that promotes scar protection [48].

The molecular signalling pathways following SCI are complex, combinatorial and densely interrelated. Each molecule can influence one or more cells and even one or two molecules coordinating with each other to elicit specific responses. Many molecules have been identified but are still under consideration to understand their mechanism regarding multiple signal regulations and multicellular interactions. Molecules associated with controlling cell death, i.e., necrosis and apoptosis, fall in different categories such as neurotransmitters, cytokines, chemokines, neuroimmune-regulators molecules (NI-Regs) and danger-associated molecular patterns (DAMPs) (Figure 5) [50].

These molecules control reactive gliosis and phagocytosis to eliminate cellular toxins. The severity of injury is defined by the releasing of these molecules; mild injury causes the releasing of extracellular glutamate and decreasing of ATP concentrations, which in return activates inflammatory responses. Higher depletion of ATP leads to an increase of cytosolic Ca^2+^ level. Macrophage activation is crucially controlled by extracellular ATP and associated purinergic signalling through connexin 43-dependent ATP release [51] (Figure 5).

The apoptosis and necrosis are further promoted by the releasing of several molecules such as DAMPs, alarmins, heat shock proteins ab-crystallin, calcium-binding protein S100, DNA binding high mobility group box 1 (HMGB1) [50] (Figure 5). The stimulation of these molecules activates the immune response and promotes the clearance of cellular debris through the stimulation of pattern recognition receptors (PRRs) [52,53] (Figure 5). The secreted alarmins then bind with PPRs to further promote the phagocytosis [52] (Figure 5). The activation of HMGBI then further promotes the activation of multiple molecules such as TLRs, receptors for advanced glycation end products (RAGE), and macrophage antigen complex-1 (MAC1). Simultaneously, the release of NI-Regs and self-defence proteins also initiates proinflammatory signals and phagocytosis [52] (Figure 5).

Another reactive molecule CD47 operates via receptors CD200R and SIRP-α present on the surface of the inflammatory cells to stop the phagocytic attack (Figure 5). Thrombomodulin (CD141) combines with HMGBI and decreases alarmin availability [53]. These multimolecular receptor-mediated signals promote reactive gliosis and cellular damage. Hence, innate immunity and adaptive immune mechanisms must be balanced. Innate immune mechanisms remove cell debris, and adaptive immune responses control molecular signalling. Reactive gliosis takes place through different adaptive immune responses and molecular signalling such as secretion of PAMPs (Pathogen-associated molecular pattern molecules), the release of liposaccharides (LPS) that skews transcriptome of reactive astrocytes toward chemokines, controlling the cytotoxicity and inflammation and balancing the coordinated multicellular innate and adaptive immune responses [54]. Interleukin 1 beta (IL-1β) serves as an important parameter that regulates the permeability of leukocytes during reactive gliosis. DAMPs and PAMPs induce the release of IL-1β that in return facilitates the release of VEGF and NG2-OPCs which further cause the release of MMP-9. This series of activation cascades promotes the transportation of serum proteins (IgGs), signalling proteins (thrombin, albumin, proteases) and leukocytes towards gliosis scar (Figure 5, Table 1) [55].

Astrocyte scar formation is driven by proliferation signals released by serum proteins and cells, such as thrombin, endothelin, FGF2, ATP, bone morphogenic proteins (BMPs), and sonic hedgehog (SHH) [41,43]. The location of scar formation is regulated from proliferating astroglial cells, fibroblast-linage cells, and inflammatory cells [56]. The scar size is controlled by the interaction of Signal transducer and activator of transcription 3 (STAT3), Suppressors of Cytokine Signaling-3 (SOCS3), or Nuclear factor-kappa B (NF-kB), and the organisation of the astrocyte scar is accomplished by the IL-6 receptor-STAT3 signalling system [57,58]. A summary of different intercellular signalling molecules, their origin and role is presented in Table 1.

### 1.4. Mechanism of Spinal Cord Recovery Pathways

The SCI causes motor and sensory dysfunctions because of the cascade of damaging events. The cascade of primary damage leads to a complex cascade of the secondary damaging events, which explains why many treatment strategies and approaches that have been studied previously were not successful in treating SCI. The available therapeutic approaches are broadly classified as neuroprotective, neuro-regenerative, and immune-modulating pathways that are briefly discussed in this section.

#### 1.4.1. Neuroprotective Pathways

Neuroprotection protects neuronal structure and function from further damage and is the relative preservation of the neurodegenerative effects of neurons and the maintenance of neuronal integrity to decrease neuronal lost ratio over time [58]. This approach prevents the progression of disease and injury from one neuron to another. Hence, neuroprotectors can be stated as disease-modifying agents that delay and even stop neuron from further degeneration [58]. The available strategies of neuroprotection can be divided into three main approaches, (i) pharmacological approaches, (ii) non-pharmacological approaches and (iii) cellular and genetic approaches.

#### 1.4.2. Pharmacological Approaches

Pharmacological approaches include neuroprotection by drugs and therapeutic agents and can be divided into different subgroups depending on the cascade of degenerative events that are being modulated. The subgroups include (i) neurotransmitter agonist and antagonist, (ii) channel blockers, (iii) anti-oxidative agents, (iv) anti apoptotic agents and (v) herbal and natural agents (Figure 6).

##### Neurotransmitter Agonist and Receptor Antagonist

Alpha 2-adrenergic (A2a) agonists prevent the neurological loss following SCI and have an imidazole ring that can interact with imidazole receptors. Their neuroprotective properties are granted by their ability to suppress the release of norepinephrine and to activate the MAPK protein kinase, the activation of which will inhibit cyclic ATP phosphorylation [59]. A2a adenosine receptor agonist ATL146e protects against tissue destruction and locomotor dysfunction post-SCI. In the rabbit model, ATL146e induced remarkable enhancement in locomotor function and neuronal viability after injury [60]. Similarly, a study in mice showed neuroprotection activity by A2a receptor agonist CGS21680, A2a receptor agonists ATL 313 and CGS 21680, which promote neuroprotection in the mouse by retarding tissue damage caused by neuronal apoptosis [60]. The release of tissue growth factors (TGF)-beta during SCI is also promoted by the activation of A1 and A2 receptors. The regulation of these adenosine receptors controls pro-inflammatory signals and responses [60]. Caffeine, an adenosine receptor antagonist, blocks adenosine receptors (A1 and A2) and provides neuroprotection against tissue damage and locomotor dysfunction [60]. Hence, adenosine receptor agonists and antagonists possess neuroprotective activity but have different mechanisms. A2a antagonist blocks excitotoxicity by reducing neurotransmitter release, whereas A2a agonist improves cell viability and motor function [60]. The ligand-gated inotropic glutamate receptors NMDA, AMPA, and kainate regulates the entry of Ca^2+^, Na^+^ and K^+^. Ca^2+^ concentration changes as a second messenger to activate intracellular SCI signalling cascades. NMDA receptors also regulate glutamate concentrations and transportations within the neurons, oligodendrocytes, and astrocytes. The high glutamate concentration causes excitotoxicity to non-injured neurons during the second phase following SCI. Therefore, NMDA, AMPA and kainate receptor antagonists aid to overcome the detrimental effect of glutamate toxicity. NMDA, AMPA, and kainate receptors antagonist memantine lead to inhibition of hypoxia, excitotoxicity, and necrosis and aids in the control of secondary injury damage [61]. Some glutamate receptor agonists/antagonists [62,63,64,65] are summarised in Table 2.

##### Channel Blockers

Na^+^ channel blockers promote neuroprotection by retarding cellular swelling, enhancing ATP loss, and improving the membrane integrity. The sodium channel blockers stop the cellular destruction by inhibiting depolarisation, cellular sodium load, and releasing the higher glutamate from neurons. Glutamate activation triggers cellular events which promote the death of neurons post-SCI; thus, its prevention could stop cellular death. Similarly, calcium signalling plays a major role in the survival of neurons. After SCI, the disrupted calcium homeostasis leads to neuronal dysfunction. Hence, the modulation of calcium within damaged tissues helps prevent neuro-degeneration [66]. The voltage-gated calcium channel blockers (VGCCs) play a vital role in calcium load regulation during SCI. VGCCs have six subtypes, i.e., L-, N-, P-, Q-, R- and T-type channels [67]. T-type calcium channels are present on neuron surface and their blockage results in long-term neuroprotection and maintenance of homeostasis by improving neuronal microcirculation [66]. L-type VGCCs include dihydropyridines such as nimodipine. N-, P-, Q- and R-type VGCCs can be blocked by several snails and spider toxins [67]. T-type VGCCs can be blocked by mibefradil with 10–30 times higher potency than nimodipine [67]. Some Na^+^ and Ca^2+^ channel blockers possessing neuroprotective activity [66,67,68,69,70,71,72] are listed in Table 3.

##### Anti-Oxidative Therapies

Anti-Oxidative Pathway: oxidative stress destroys proteins, lipids and DNA by producing ROS and RNS in the brain and spinal cord [73]. ROS and RNS production increase ascorbic acid demand and alters the ability of antioxidant enzymes such as superoxide dismutase (SODs), the catalase, and the glutathione [73]. The Nrf2 signalling pathway (nuclear factor E2) is the main cause of cellular defence against oxidative stress. The Nrf2 activates phase II detoxifying enzymes via antioxidant response element (ARE) regulation [74]. Antioxidant response element (ARE) also activates NF-kappa B inflammatory responses [69].

Anti-Oxidant Approaches: The mechanism of ROS and RNS production has been discussed above. The production of reactive oxygen and nitrogen species during injury produces the oxidative stress on healthy neurons and potentiates further neuro-degeneration. The antioxidants are chemical moiety that prevents the body from oxidative stress by inhibiting the oxidation of different molecules [75]. Hence, ROS and RNS inhibitor counteract the oxidation of various bioactive molecules that take place during the secondary phase of spinal injury. Numerous molecules are being used to control ROS and RNS generation. The antioxidant therapies are categorised into two therapeutic groups i.e.,: (i) the compounds inhibiting ROS and RNS generation and (ii) the compounds that inhibit lipid peroxidation (LPO) [76]. Glutathione, a tripeptide that produces glutathione monoethyl ester (GSHE) by reduction process, acts as an antioxidant in controlling apoptosis and retard ROS generation. The GSHE diminishes SC LPO generation and the glutamate excitotoxicity [74]. Omega-3 fatty acid (ω-3 PUFAs) and docosahexaenoic acid (DHA) possess anti-inflammatory, antioxidant and membrane-stabilising activity. ω-3 PUFAs and DHA act on cyclooxygenase (COX) pathways, cytosolic phospholipase A2 (cPLA2), and kappa-light-chain-enhancer (NF-kB) and inhibit the production of ROS, RNS and lipid peroxidation of nerve cells [74], promoting neuroprotective pathways (Figure 6).

The glucocorticoids such as dexamethasone and methylprednisolone (MP) are being used for SCI treatment for years. Glucocorticoids act on neuron excitability, inhibit LPO, and ROS formation. However, the guidelines provided by American Association of Neurological Surgeons (AANS) and Congress of Neurological Surgeons (CNS) in 2013, proposed the limitations for administration of corticosteroids at a level I recommendation only (treatment strategies supported by Class I medical evidence), because of several reasons (i) various corticosteroids are not recommended by Food and Drug Administration (FDA), (ii) no clinical trial Class I or Class II evidence support a clinical benefit, and (iii) clinical trial class I, II, and III evidence indicate that the high-dose of corticosteroids are associated with harmful side effects including death [77]. MP treatment in the cat model inhibits ROS and RNS generation and LPO level when given intravenously [76]. MP possesses neuroprotective activity because it inhibits ischemia, promotes aerobic metabolism, reduces calcium overload, and inhibits calpain-dependent neurotoxicity [76]. A high dose of MP causes LPO inhibition, whereas low doses promote anti-inflammatory and anti-oxidative activity. Although high doses of MP for acute SCI responses are previously recommended, new guideline by AANS and CNS (2013) has restricted MP use for acute SCI because this drug showed modest efficacy but also possible severe complications [74,77]. High-dose MP therapy is no longer routinely used in acute SCI but remains an optional therapeutic approach in certain conditions [74]. A recent guideline restricts the administration of 24 h infusion of high-dose MP within 8 h of acute SCI as a treatment option and did not recommend 48 h infusion of high-dose MP [77]. [Several anti-oxidative agents such as polyethyleneglycol-conjugated-SOD (PEG-SOD), tirilazad, and dexanabinol show only minimal neuroprotective activity [78]. Tirilazad inhibits LPO generation through membrane stabilisation and scavenging (Figure 6).

Several new antioxidants that promote neuroprotection have been found in recent studies. U-83836E(2-[[4-(2,6-dipyrrolidin-1-ylpyrimidin-4-yl)piperazin-1-yl]methyl]-2,5,7,8-tetramethyl-3,4-dihydrochromen-6-ol dihydrochloride) is a second-generation of lazaroid (a class of lipophilic steroids that inhibits LPO), containing a non-steroidal structure and an α-tocopherol ring [79]. U-83836E was shown to inhibit LPO, ROS, and RNS production. U-83836E inhibits calpain-dependent neurodegeneration and cascading events associated with secondary injury pathways and acts as a neuroprotective agent [79]. Another drug melatonin (N-acetyl-5-methoxytryptophan) scavenges free radicals (ROS and RNS) and regulates endogenous antioxidant enzyme expressions [78]. Melatonin also decreases LPO, preserves neuronal structures, and increases neuroprotection post-injury. Melatonin and dexamethasone combination showed good neuroprotective activities by acting as an anti-inflammatory agent and improving locomotor function [76]. This drug compound improves the brain anti-oxidant level, reduces NF-kappa B activation and enhances cognitive function in traumatic brain injury (TBI) models [79]. Other drugs such as penicillamine and phenelzine promote sensitivity to LPO-derived aldehydes that act as carbonyl scavengers and shown to improve neuronal function and neuroprotection in concussive mouse injury model [79]. Nitroxide-containing antioxidants, such as tempol, it acts as a potent antioxidant that retards ROS and RNS formation [80]. In the mouse model, tempol inhibits LPO and protein nitration and consequently neuronal oxidative stress, reduces calpain-mediated neuro-degeneration [75] and brain oedema post-trauma and promotes locomotor function recovery in rats [80]. However, the exact mechanism of action of tempol must be further investigated. Resveratrol, a polyphenolic drug, also has neuroprotective activity in neurotraumatic stroke and Alzheimer’s disease. Resveratrol decreases oxidative stress, post-SCI oedema, Na^+^, K^+^-ATPase activity and improves neurological activity during SCI [81]. Resveratrol decreases malondialdehyde (MDA) expression and superoxide dismutase (SOD) activity, inflammatory cytokines, xanthine oxidase activity and apoptotic protein activity and promotes neuroprotective activity [82]. Nrf2/ARE signalling activators are potent antioxidants. Nrf2 (nuclear erythroid 2-related factor 2) is a transcription factor that is attached to the ARE and regulates gene expression which is included in the cellular antioxidant and anti-inflammatory defence mechanisms along with mitochondrial protection [82]. Sulforaphrane, an Nrf2/ARE signalling activator, reduces oedema, decreases glutamate concentration, and reduces inflammatory cytokines IL-1β and TBFα activity [83]. The treatment using sulforaphane in the mouse model promotes the expression of Nrf2 and glutathione S-transferase- α1 (GST-α1) [81]. Another Nrf2/ARE activator, tert-butylhydroquinone decreases the neurological oedema, neuro-inflammation, NF-KB activation, and TNFα and IL-1β formation. Thus, in return it retards the oxidative stress and neurotoxicity [84].

##### Apoptosis-Related Signaling Pathways Inhibitors

Apoptotic Pathways: The apoptotic pathways are further divided into two major pathways, i.e., (i) the death receptor initiated (also called extrinsic) pathway and (ii) the mitochondrial (also called intrinsic or Bcl-2-regulated) pathway (Figure 7). These pathways are initiated by stimulation of the caspases (cysteine-associated aspartate proteases), that act as a vital component of the programmed cell death. The caspases are also considered as initiators (caspases 2, 8, 9 and 10), executioners (caspases 3, 6 and 7) and inflammatory caspases (caspases 1, 4 and 5) respectively. Few other caspases, i.e., 11, 12, 13 and 14 have been identified as specific apoptotic agents [85]. The extrinsic pathways also termed as death receptor pathways are mediated by TNFR (tumour necrosis factor receptor), Fas, and tumour necrosis factor (TNF)-related apoptosis-inducing ligand (TRAIL) [86]. The activation of death receptors causes recruitment and activation of caspases 8 and 10, which trigger the procaspase-3 activity that consequently promotes the conversion of caspase 3 (Figure 6). The increasing in glutamate and MPP+ stimulation post-SCI leads to enhance the release of cytochrome-C and pro-apoptotic proteins [85,86]. The release of cytochrome-C initiates the intrinsic pathway through recruitment of apoptotic protease activating factor-1 (Apaf-1) and pro-caspase-9 activity which motivates the formation of apoptosome [87]. The newly generated apoptosome activates and regulates the signalling cascade from caspase-9 to caspase-3 and then causes apoptotic cell death. Extrinsic and intrinsic pathways follow different steps but both include the activation of caspase-3 regulating cell death [88] and are controlled by the activity of several proteins such as glycogen synthase kinase-3 (GSK3), ataxia telangiectasia, mutated (ATM)/p53 (a nuclear transcription factor), B-cell lymphoma-2 (Bcl-2), cyclin-dependent kinases (CDKs) and MAPKs. ROS production is a sensor promoting DNA damage. Another sensor that also promotes the DNA damage is ATM, a member of the PI3K family [87,89]. The stimulation of ATM contributes to the activation of p53, which subsequently produces an apoptotic signal to mitochondria by intrinsic pathways. The continuous damage to DNA causes the over-activation of p53, which increases the expression and mediation of BH3 (pro-apoptotic) and PUMA (p53 upregulated mediator of apoptosis) [87]. The increase in BH3 and PUMA expression further activates the Noxa pathway. Noxa is a pro-apoptotic gene belonging to the Bcl2 protein family that contains the BH3 domain and is another contributor for apoptotic pathways [87].

PUMA binds to Bax and Bcl-2 family members (Bcl-2, Bcl-XL, Bcl-w, and Mcl-1). The Bcl family proteins can be classified as: (i) pro-apoptotic proteins (Bax and Bak) and (ii) anti-apoptotic proteins (Bcl-2 and Bcl-XL) [88]. The pro-apoptotic proteins regulate the permeability of the mitochondrial membrane through dimerisation and oligomerisation of voltage-gated anionic channels (VDAC) [87]. The Bax and Bak permeability into the mitochondrial membrane promotes cytochrome-c release which initiates apoptosis. Under normal conditions, the balance ensues between the pro-apoptotic protein and the anti-apoptotic protein. However, during injury state, this balance is disturbed, promoting the generation of BH3 interacting-domain death agonist (Bid) proteins () [85]. These Bid proteins act as a link between extrinsic and intrinsic pathways which are regulated by caspase-8 activity amplifying death signals. Bid proteins also interact with Bax and Bak proteins. The movement of Baxs and Baks is controlled by these proteins through mitochondrial VDAC [86]. However, the Bcl-2 and Bcl-XL anti-apoptotic proteins retard Bax movement to mitochondria and bind to Apaf-1. These binding complexes then retard the caspase-9 initiation. The poly-ADP-ribose polymerase-1 (PARP-1) activity is initiated by glutamate excitotoxicity; inhibition of glutamate excitotoxicity further stops ROS generation. The PARP-1 also binds to the NMDA receptor and prevents mitochondrial damage [86] (Figure 7).

Caspase Inhibitor: Z-DEVD-fmk (the peptide N-benzyloxycarbonyl-Asp(OMe)-Glu(OMe)-Val-Asp(OMe)-fluoro-methyl ketone) is a selective caspase-3 inhibitor that also possesses anti-inflammatory properties. Anti-apoptotic agents usually block apoptosis and cytokine production; reduces tissue destruction and ischemia, restores locomotor activity; and enhances neuroprotection [87]. When introduced within 30 min of optic nerve injury in a rabbit model, Z-DEVD-fmk establishes neuroprotective property by reducing apoptosis [89]. The z-LEHD-fmk (caspase inhibitor) incorporates anti-apoptotic properties, and the neuroprotective properties [90]. However, caspase inhibitors could only temporarily improve the neuronal function and neuroprotection; therefore, the usage of caspase inhibitor alone is not considered as a successful neuroprotective strategy [85]. Tetrapeptidyl chloromethyl ketone (Ac-DEVD-CMK) is also a caspase 3-inhibitor, which blocks caspase 3-dependent apoptotic pathways and exhibit neuroprotection properties [91] (Figure 6 and Figure 7).

Calpain Inhibitors: The two types of calpain involved in SCI are µ- and m-calpain, both containing an 80 kDa catalytic subunit and are encoded by genes calpain-1 catalytic subunit (CAPN1) at chromosome 11 and calpain-2 catalytic subunit (CAPN2) at chromosome 1. The main difference between these two calpains depends on varying calcium concentrations; the m-calpain has calcium amount range between 3–50 mM, while µ-calpain contains calcium amount ranges between 0.4–0.8 mM [92,93]. The cysteine proteases along with the calpain increases immunoreactivity by promoting the neurofibrillary pathology and cause the synapse loss and apoptosis. Thus, calpain inhibitors can stop neuronal loss. Cysteic leucyl argininal (CYLA), is a calpain antagonist that inhibits retinal ischemia and apoptosis by decreasing the glutamate excitotoxicity in the rat model [94]. The irreversible cysteine protease inhibitor E-64-d inhibits the apoptosis following SCI. In a study done on rat model E-64-d prevented the calpain-mediated neuronal apoptosis in the core lesion formed during SCI. The E-64-d prevents the calpain 1 activation and COX-2 activity and in return ceases the neuronal apoptosis through inhibiting the stimulation of caspase-3, AIF is released and thus improves the locomotor recovery in SCI [95] (Figure 6 and Figure 7). Another calpain inhibitor named calpastatin can also improve neuroprotection by inhibiting calpain-associated apoptosis [93].

Other Anti-Apoptotic Agents: Glycogen synthase kinase-3 (GSK-3) inhibitor performs a vital part in the regulation of apoptotic intracellular signal pathways. 4-Benzyl-2-methyl-1, 2, 4-thiadiazolidine-3, 5-dione (TDZD-8), a GSK-3 inhibitor, inhibited neuronal apoptosis, GAP-43 expression, and increased locomotor function and recovery in SCI [96]. Neuronal cell division cycle is controlled by serine/threonine kinases (CDks), and during injury cyclin A and E2F-1 expression altered and resulted in neuronal apoptosis. The G1 and S phase inhibitors such as flavopiridol, kempaullone, and roscovitine possess neuroprotective activity by blocking the formation of ROS and RNS and hence stop apoptosis [84]. 3-Hydroxy-3-methylglutaryl-CoA (HMG-CoA) reductase inhibitors (statins) are known to have anti-inflammatory and neuroprotective properties, and the reported mechanism involves the inhibition of ROS generation and neuroprotection against glutamate excitotoxicity [97,98]. The statins have been demonstrated as neuro-protectants, which prevent glutamate excitotoxicity. Some commonly used statin such as simvastatin, lovastatin, fluvastatin, pravastatin, and atorvastatin retard the neuronal damage [97]. Among all commercially available statins, simvastatin is found to be more effective neuroprotective agents that can reduce glutamate excitotoxicity, stop oxidative damage, and inhibit neuritic dystrophy in return prevent from apoptosis following SCI [98]. Lovastatin is the second most effective statin in preventing glutamate excitotoxicity. While other statins are found to possess fewer neuroprotective activity in the mouse model [97].

##### Herbal and Natural Agents

Many natural constituents such as polyphenols, phenolic acids, curcuminoids, resveratrol, flavonoids, alkaloids, and terpenoids show neuroprotective activity. Many polyphenols control biological activities such as chromatin remodelling and epigenetic modifications [99]. Phenolic compounds such as rosmarinic acid, flavonoids, ferulic, chlorogenic, caffeic, vanillic, p-hydroxybenzoic acid, protocatechuic acid, and p-coumaric acid are antioxidant agents. These agents could modulate the hydrogen transportation, electron donation, free radical scavenging, metal chelation, alteration of antioxidant levels, and activation of enzymes and regulation of Nrf2 pathways [100]. Some of the commonly used natural neuroprotective agents [100,101,102,103,104] are listed in Table 4.

#### 1.4.3. Non-Pharmacological Approaches

The non-pharmacological approaches include vitamins, growth factors, and cultured cells. The non-pharmacological approaches may contribute to effectively reduce SCI complications such as pain, swelling, and improve locomotor activity by utilising non-medication approaches. These non-pharmacological approaches are beneficial for short duration and for long-term clinical efficacy they should be combined with pharmacological agents [106]. Therefore, prevention and treatment of ischemic brain injury require multiple interventions. Further study is needed for the effective outcome of non-pharmacological approaches, particularly those with few side effects [106]. Natural vitamins attack generation of ROS and RNS that further retard LPO and cellular damage. The vitamins such as vitamins A, E, and C are natural antioxidants. Vitamin A enhances the release of IL-1β, IL-6, and TNFα, improving neuroprotection [107]. Vitamin C retards lipid hydroperoxides formation and stops membrane destruction. Several other neuroprotective pathways are demonstrated such as (i) diminish the necrotic tissues and promotes functional recovery, (ii) retards ROS, and LPO generation, (iii) reduces the expressions of proteins such as NF-kB, iNOS, and COX-2, (iv) down-regulates the levels of TNFα and IL-1β, and (v) controls antioxidant status and MPO activity [105]. Vitamin E increases functional recovery by reducing ROS, RNS, LPO, glutathione activity, and it also reduces peroxidases [108]. Resveratrol is a natural phytoalexin exhibiting neuroprotective activity that prevents oedema formation, glutamate excitotoxicity and neuro-regeneration [108]. Selenium promotes neuroprotective activity against oxidative stress accompanying SCI [109]. Glutathione peroxidase (GPx) and thioredoxin reductase (TrRx) contain selenium; therefore, selenium possesses antioxidative activity and prevents the oxidative stress associated with ROS production [109]. Coenzyme Q10 (CoQ10) inhibits the mitochondrial dysfunction by retarding higher ATP synthesis, decreasing ROS formation, and reducing the neurodegenerative stress [108]. Other approaches for neuroprotection include therapeutic hypothermia, which decrease metabolic rate and inhibit inflammatory responses [110] (Figure 6). Surgical decompression has shown potential advantages, by promoting neurological recovery and preventing further neurological deterioration following secondary injury. Surgical decompression performed within 48 h post-injury reduces pressure, further protects the spinal cord [110], diminishes progressive oedema and haemorrhage after SCI and decreases the pressure caused by oedema and inflammatory responses; therefore, patients who undergoing surgical decompression have a good chance of recovery [111].

#### 1.4.4. Cellular and Genetic Approaches

Other cellular approaches are growth factors including brain-derived neurotrophic factor (BDNF), transforming growth factor-β (TGF-β), and insulin-like growth factor-1 (IGF-1) which act as neuroprotective agents. Granulocyte colony-stimulating factor (G-CSF) inhibits glutamate excitotoxicity, apoptosis, and activation of TNF-α and IL-1β [112]. BDNF improves functional recovery, antioxidant stress, neuronal survival, and neuroprotection against TBI [113]. Transforming growth factor-beta (TGF-β) promotes neuronal differentiation, migration and neuroprotection. TGF-β is given post-SCI to elevate the immune response, induce the formation of glial scar and promote functional recovery [114]. Stem cell therapies are innovative approach that may solve the challenges in SCI treatment because of their neuro-regenerative, neuroprotective and immunomodulatory properties [115]. Stem cell therapies such as neural stem cells (NSCs), bone marrow stem cells (BMSCs), olfactory ensheathing cells (OECs) and Schwann cells (SCs) are gaining popularity [116]. NSCs reduce neutrophils and M1 macrophages; down-regulate TNFα, IL-1 β, IL-6 and IL-12; improve functional recovery; and decrease apoptosis and microglial activation, thus improving locomotor and sensory functions [116]. BMSCs improve tissue protection and locomotor function, increase neurotropic growth factor, activate M2 macrophages and inhibit glial scar formation [117]. OECs reduce scar size and increase neurofilament sprouting and axon functions [118]. SCs up-regulate NOS expression, activate the c-GMP pathway, stimulate neuronal growth factor BDNF expression and reduce inflammatory cytokines and ROS formation, thereby promoting neuroprotection [119] (Figure 6). Other promising cellular therapies include utilisation of induced pluripotent stem cells (iPSC) and ependymal stem/progenitor cells (epSPC) for treatment of SCI [115]. Future studies must focus on the cellular therapies as ideal approaches for SCI treatment as stem cell-based therapy have been proven safe, with the aim to utterly exploit the promising therapeutic potential of both exogenous and endogenous stem cells in SCI [115].

Biomaterials also possess the potential as a therapeutic option for SCI treatment. Several bioengineering technologies have been considered, but many of them are still in preclinical stages and in vitro stage of investigation [120]. The commonly reported biomaterials for treatment of SCI includes are phase-separated poly (2-hydroxyethyl methacrylate) (pHEMA), alginate, HPMA copolymers, hyaluronic acid, agarose/carbomer hydrogels, BD puramatrix synthetic peptide, and collagen [120]. The bioengineered approach used for SCI treatment can be formulated as sheets, hydrogels, scaffolds, nanoparticles, nanofibers and magnetic microgels [120]. Bioengineered therapies benefit the release of administered drug or cells to the host tissues and thus promote neural regeneration. Several biodegradable biomaterials such as poly lactic co-glycolic acid (PLGA), poly-l-lysine (PLL), poly ethylene glycol (PEG), poly vinyl alcohol (PVA), poly(ɛ-caprolactone) (PCL), gelatin, fibrin, laminin and grapheme have been used to produce hydrogels, nanoparticles, scaffolds, nanogels and magnetic nanofibers and were incorporated with single or multiple therapeutic agents to promote neuroprotection, immuno-modulation and neuro-regeneration [121]. Bioengineered therapeutic approaches benefit the delivery of therapeutic agents and cells and support the survival of delivered cells in situ [120,121].

#### 1.4.5. Immuno-Modulatory Pathways

##### Neuroinflammation

Neuroinflammation is a vital component of secondary responses after neuronal injury. Inflammation is previously referred to as a detrimental outcome of injury. In SCI, inflammation can either be beneficial or destructive [116]. Neuroinflammation consists of multicellular interactions and cells which are involved in inflammatory reactions such as neutrophils, resident microglia, astrocytes, dendritic cells (DCs), blood-borne macrophages and B- and T-lymphocytes [122]. Neuroinflammation occurs in stages, the first phase of inflammation involves migration of resident microglia, astrocytes and neutrophils towards an injured site. The second phase involves the migration of blood-borne macrophages, B and T lymphocytes towards the injured area. B cells produce autoantibodies that introduce neuroinflammation and tissue destruction [123]. Each immune cell plays a unique role and has a unique interaction with each other as discussed in the next section.

Astrocytes: are not immune cells but perform a pivotal contribution in the neuro-inflammatory pathway [124]. Astrocytes usually regulate homeostasis, serve nutrients and growth factors to neuronal tissues, regulate glutamate transport, and eradicate excessive fluids and ions [124]. Astrocytes regulate adaptive and innate immune responses by promoting differential signalling pathways. Astrocyte regulates cytokines and chemokines production, recruit neutrophils production by IL- 1R1-Myd88 pathway, NF- kB pathway, and control expression of ICAM (intracellular adhesion molecule) and VCAM (vascular cell adhesion molecule) [125]. Post-injury IL-1β production increases in astrocytes and microglia, which in turn increases expression of monocyte chemo-attractant proteins (MCP)-1, chemokine C-C motif ligand 2 (CCL2), C-X-C motif ligand 1 (CXCL1) and C-X-C motif ligand 2 (CXCL2) [122,123]. Astrocytes also enhance M1 and M2 pro-inflammatory chemokines production through expression of TNF-α, IL-12, and IFN-γ and anti-inflammatory cytokines TGF-β and IL-10. IL-6 cytokine gp130 activates SHP2/Ras/Erk signalling while TGF- β signalling inhibits NF- kB activity [125]. Astrocytes also regulate the STAT3 signalling pathway, promote STAT3 phosphorylation increase scar formation, and restrict inflammation (Figure 7). Astrocytes also produce IL-17R and stimulation of NF-κB by IL-17R which promote pro-inflammatory mediators, oxidative pathways, and neuroinflammation [126].

Neutrophils: neutrophils migrate towards the injured site within 24 h and this contributes towards phagocytosis and clearance of cellular debris [127]. Neutrophils control inflammatory cytokines, proteases and free radicals, activate astrocytes and microglia, and control neuroinflammation [125]. Neutrophils regulate specific antibody LyG6/Gr1+, regulate IL-1 receptor antagonists, and promote neuroprotective activity [128]. Neutrophils depletion result in a decrease of cytokines and chemokines expression, down-regulation of fibroblast growth factors, vascular endothelial growth factors (VEGFs), and morphogenetic proteins (BMPs) and halt normal healing mechanism [128].

Microglia: microglia is resident immune and macrophage cells [129]. Upon injury, monocyte infiltrates spinal cord tissue converts into macrophages [129,130]. Macrophages and microglia promote neuro-regeneration by regulating growth factors such as nerve growth factors (NGF), neurotrophin-3(NT-3), and thrombospodin [131]. They also promote phagocytosis and scavenge damaged spinal tissue, and clear myelin debris. Microglia and macrophages act as M1-like (pro-inflammatory) and M2-like (anti-inflammatory pro-regenerative) phenotype [131]. M1 like phenotype induces Th1 specific cytokines, interferon (IFN)-γ, TNF-α, and intracellular accumulation of iron [132]. M1 like microglia/macrophages express MHCII and promote antigen for T cells and activate, phagocytosis, innate, and adaptive immune responses [132]. M2 like phenotype is polarised by Th2 cytokines, IL-4 and IL-13, and reduces NF-κB pathway activity. IL-4 delayed the expression of M2 markers in microglia and macrophages; hence the delayed administration of IL-4 (48h after SCI) markedly improves the functional outcomes and reduces the tissue damage after contusion injury [133]. IL-10 is an immune-regulatory cytokines that promotes tissue repair and regeneration, IL-10 mediate phagocytosis, and oligodendrocytes differentiation [134], M1 fabricate macrophages responses while, M2 promotes fibrotic scar formation via the release of specific factors such as TGF-β, PDGF, VEGF, IGF-1 and galectin-3 [134,135].

T and B lymphocytes: play a pivotal role in adaptive immune responses, adopt different phenotypes and contribute to injury and repair processes. T cells induce detrimental effects on neurons and glial cells. Teff cells control neuronal cell function by regulating the production of several pro-inflammatory cytokines and chemokines such as IL-1β, TNF-α, IL-12, CCL2, CCL5, and CXCL10 [136]. Treg cells on the other hand control release of anti-inflammatory cytokines IL-10 and TGF-β. Treg cells also regulate Teff cell activation during normal neuronal functioning; however, during SCI, the Teff and Treg balancing regulation got interrupted causing more activities of Teff cells, resulting in the higher release of pro-inflammatory cytokines and chemokines causing the enhancement of Fas-mediated apoptosis [137]. These autoreactive Teff cells promote differentiation of B lymphocytes into autoantibody, which further potentiates neuronal apoptosis [137].

Post SCI the activation of B cells is suppressed, which leads to suppression of antibody production [131], but according to a study, the activity of B cells got influenced by the levels of injury, upper thoracic SCI retard antibody production however mid-thoracic injury produces no effect on antibody production. An increase in serum corticosterone and norepinephrine level cause suppression of B cells promoting lymphocyte apoptosis [138]. Despite producing detrimental effects B cells also contribute to spinal cord repairing post-injury by regulating autoimmune responses [139]. B cells contribute to SCI repair by producing immunomodulatory Breg phenotype, which regulates T cells autoimmune responses by controlling IL-10 production (Figure 8) [139].

##### Immunosuppressive or Immunomodulatory Drugs

Immunomodulatory drugs usually alter the response of immune cells either by immune-stimulators and immune-suppressive activity. Immuno-stimulators usually promote immune responses during various disease states (infectious diseases and tumours), whereas immune-suppressive drugs reduce the immune responses after SCI [140]. Some of the commonly reported immunosuppressive drugs [139,140,141,142,143] used in SCI are listed in Table 5.

#### 1.4.6. Neuro-Regenerative Pathways

Neuro-regeneration is the regrowth and repair of damaged nervous tissues (neurons, axons, synapses and glial cells) after injury. Neuro-regeneration includes either the elongation of axons, sprouting and and growth of new axons or the remyelination of nerve cells. The neuro-regeneration approaches can be divided into two main types i.e.,: (i) stimulation of axonal sprouting and growth and (ii) inhibition of glycoprotein and proteoglycans (Rho-ROCK pathway) [144].

##### RhoA-ROCK Kinase Pathway

RhoA is a small GTPase protein belonging to the Rho GTPase family. RhoA downstream effector (ROCK) regulates the neuronal cytoskeleton. ROCK 1 and ROCK 2 pathway controls cell contraction, motility, proliferation, gene expression and apoptosis [145] and regulates inflammatory responses and mediates inflammatory cell infiltration and migration. RhoA/Rho kinases regulate neuroinflammation, neuropathic pains and apoptosis during SCI [145] and the production of cytokines such as necrosis factor-α (TNF-α), interleukin-1 beta (IL-1β), interleukin-2 (IL-2) and CXC chemokines [146]. ROCK reduces leukocyte infiltration, cytokine production and lymphocyte proliferation.

The RhoA/Rho pathway controls the three important events associated with SCI: the regulation of neuropathic pain, apoptotic cascade and axon degeneration. (i) Firstly, the Rho pathway controls neuropathic pain by lysophosphatidic acid which is usually found at the lesion core, initiates neuropathic pain [147], binds to G-protein coupled LPA receptors and activates RhoA/Rho pathway [148]. RhoA/Rho-kinase mediates p38 MAPK activation via morphological changes in ATP receptors that induce neuropathic pain [148]. RhoA/Rho activation promotes the production of pro-apoptotic proteins p75NTR that is responsible to activate apoptotic cascade. The decrease in p75NTR generation also decreases the apoptosis during SCI [147]. The activation of RhoA/Rho also activates p38α. The activation of p38α initiates the excitotoxic neuronal death [147]. (ii) The second important event of RhoA/Rho-kinase includes the regulation of cell death. Rho-kinase controls the myosin light chain phosphorylation and promotes actomyosin contractility, which induces cell membrane blebbing and fragmentation and simultaneously promotes neuroinflammation and ROS production, resulting in cellular apoptosis [147]. ROCK2 enhances apoptosis by increasing Fas-induced cell death. Rho-kinase activates phosphatase and tensin homologue (PTEN) and insulin receptor substrate 1 (IRS1) and promotes ROS production to induce apoptosis [146] (Figure 9). (iii) Thirdly, RhoA/ROCK pathway also prevents axon from regeneration by stimulating myelin-associated glycoprotein inhibitors such as Nogo, myelin-associated glycoprotein and oligodendrocyte myelin glycoprotein (OMgp) [148]. The myelin-associated glycoprotein is involved in axon regeneration, and its inhibition results in further axonal degeneration [146]. Hence, neuro-regenerative strategies should act on one or more events initiated by the Rho/ROCK pathway. Some of the important strategies are discussed below.

#### 1.4.7. Neuro-Regenerative Approaches

Enhancement of Remyelination: Remyelination successfully promotes the action potential and survival of axons and corresponding neurons [147]. Thus, remyelination pathway is an attractive therapeutic target for regenerative medicine for clinical trials following SCI [147]. GTPase, RhoA, activates ROCK to inhibit neurite outgrowth and neural growth. NOGO-A (myelin protein) receptor antagonists, anti-NOGO-A antibodies, or RhoA-ROCK inhibitor promote neurite growth and axonal regeneration [148]. Transplantation of stem cells such as SCs, OECs, NSCs and OPCs is the most promising strategy to promote remyelination. Cell therapies may provide neuroprotective and neuro-regenerative actions. Transplantation of Schwann cells (SCs) post-SCI promotes the generation of myelin sheaths and the production of growth factors, extracellular matrix and adhesion molecules [148]. OECs are specialised glial cells in olfactory system that promote the growth of new olfactory epithelium by lamina propria into nerve layers of the olfactory bulb. OECs secrete lipid vesicles, neurotrophic factors, and extracellular matrix molecules promote remyelination [149]. Besides that, MSCs have potent anti-inflammatory, anti-apoptotic, immunomodulatory, and angiogenic effects post-SCI [149]. Neural stem cells (NSCs) and neural progenitor cells (NPCs) can differentiate the 3 major cells of the central nervous system (CNS) such as neurons, astrocytes, and oligodendrocytes. This has made them very attractive for cell replacement therapy post-SCI, which aid in the myelination of the demyelinated axons and lead to improvements in axonal conduction [149] (Figure 9).

Enhancement of Neuronal and Axonal Regeneration: Oligodendrocytes myelin inhibitor (35 and 250 kDa) and monoclonal antibody (IN-1) prevent the inhibitory factors, which in return promote the axonal regeneration and improvement of locomotor function post-SCI. Similarly, chondroitinase ABC (ChABC), isolated from Proteus Vulgaris retards the chondroitin sulphate proteoglycan (CSPGs) production that inhibits axonal regeneration. [150]. ChABC also liberates growth factors, prevents receptor-mediated from inhibition, promotes anti-inflammatory effects and axonal sprouting and increases regeneration [151]. Self-assembling peptides (SAPs) constructed as nanofibers can minimise the damage by inhibiting inflammation, astrogliosis and neuronal apoptosis and can frequently fabricate axonal regeneration [151]. Y27632 inhibits the Rho-associated kinase (ROCK) activity, hence promotes neurite outgrowth. Simultaneously, fasudil, a ROCK inhibitor, was found to improve the functional recovery by inhibiting inflammatory responses and CSPGs secretion. Clostridium botulinum enzyme, the other Rho inhibitor C3 transferase was found to promote axon regeneration [152] (Figure 9).

## 2. Discussion

SCI is a devastating condition [1]. Substantial progress has been made in understanding the pathophysiology of spinal cord injuries; however the various therapeutic interventions have distinct advantages and limitations [1]. The first problem is how to prevent the cascade of events which is associated with the secondary spinal injury phase. The second challenge includes the regeneration of injured spinal tissue and restoration of the lost connectivity. The pathophysiology of SCI is dynamic and complex involving interrelated molecular and biochemical events [6]. Various treatments have been designed to control a single aspect of events or multiple events simultaneously [7]. Treatments regulating and controlling concomitant pathways either directly or indirectly helps improve this devastating condition. Most successful approaches act as optimal to overcome complications related to SCI [153]. Many potential therapies have shown efficacy in preclinical trials up to phase IV but some therapies have demonstrated drawbacks such as unacceptable doses or unfavourable pharmacokinetics, short half-life, and pharmacodynamics parameters. These challenges can be overcome by designing an appropriate drug-delivery systems that directly affect drug bioavailability and specificity, reduce adverse drug effects and can incorporate single or multiple drugs targeting neuroprotection and neuro-regeneration and prolonging drug effects. In addition to drugs and active compounds, various cellular and genetic approaches have shown promising effects in controlling the detrimental effects of SCI [154]. Researchers aimed to exemplify the feasibility of novel approaches, considering the cascading events that occur during pathophysiology of SCI, multicellular and multimolecular interactions and promising treatments for the neuroprotection, immunomodulation and neuro-regeneration of spinal cord. Several recent approaches were successful in eliminating or reducing detrimental effects and the combination therapy using stem cells and neuroprotective or neuro-regenerative agent’s shows potential in providing good outcomes.

## 3. Conclusions

SCI has emerged as one of the most devastating conditions with a remarkable effect on healthcare systems worldwide. Unfortunately, no permanent cure is available for SCI. Developing a combinative approach utilising neuroprotective and neuro-regenerative strategies to simultaneously target multiple pathways will be beneficial. Similarly, providing appropriate drug-delivery systems of new neuroprotective and neuro-regenerative agents or their combinations can improve the efficacy of available SCI treatments. Hence, an articulate approach must be developed to target numerous degenerative pathways and provide favourable conditions to promote repair mechanisms. The multicellular and multi-molecular interaction mechanisms of neuro-regeneration and neuroprotection must be clearly understood. Various available drugs, biologics and cell therapies must be devised into an effective combinative treatment modality for complete nerve regeneration, a complex process that needs a long period.

## Figures and Tables

**Figure 1 ijms-21-07533-f001:**
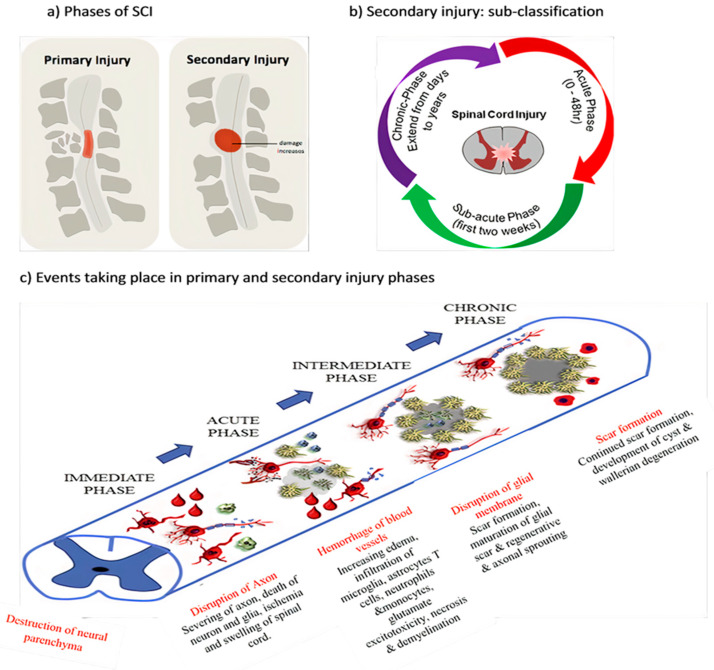
Spinal cord injury (SCI) (**a**) phases of SCI, (**b**) sub-classification of secondary injury depending on duration of injury and (**c**) pathophysiological events according to SCI phases.

**Figure 2 ijms-21-07533-f002:**
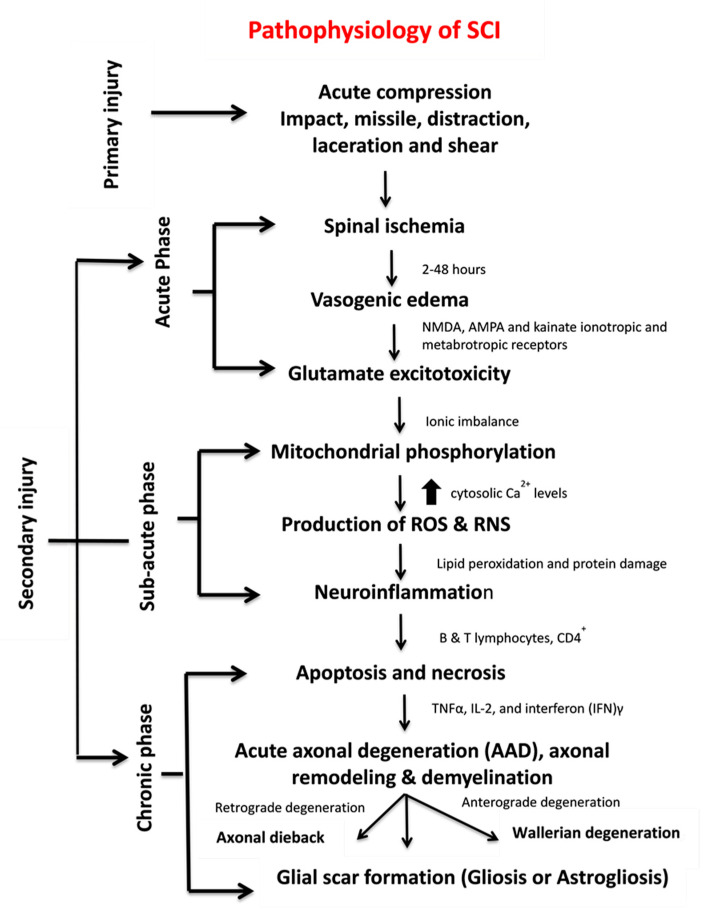
Pathophysiology, clinical manifestations, and phases of SCI.

**Figure 3 ijms-21-07533-f003:**
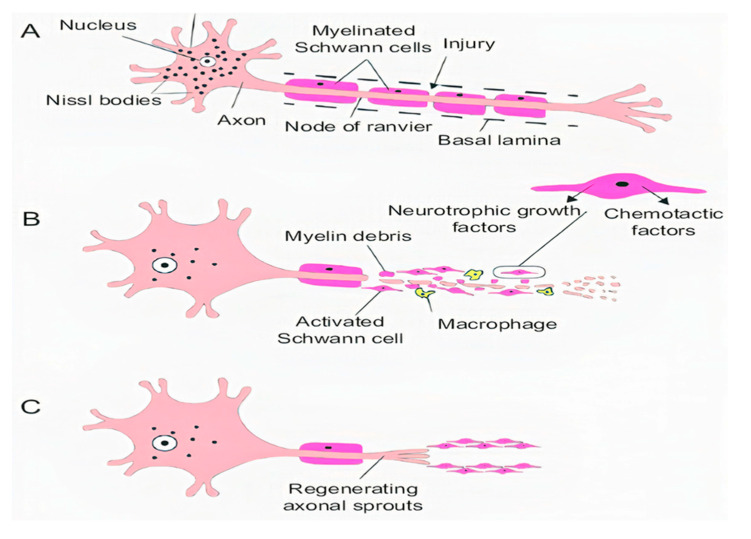
Stages of axon degeneration, (**A**) acute injury responses, (**B**) acute axonal degeneration (AAD) and (**C**) Wallerian degeneration.

**Figure 4 ijms-21-07533-f004:**
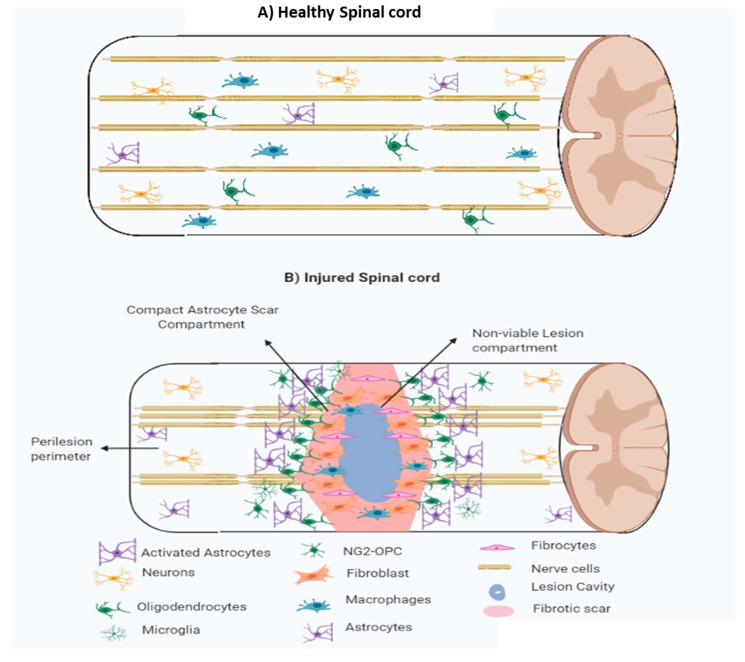
(**A**) Healthy spinal cord and (**B**) an injured spinal cord with three lesion compartments, showing inner non-viable small lesion compartment, compact astrocyte core, and perilesion perimeters with multicellular and multi-molecular components (astrocytes, neurons, macrophages, microglia, NG2-OPC, fibrocytes, oligodendrocytes, fibroblast, nerve cells and activated astrocytes) regulating gliosis (gliosis scar formation) post SCI.

**Figure 5 ijms-21-07533-f005:**
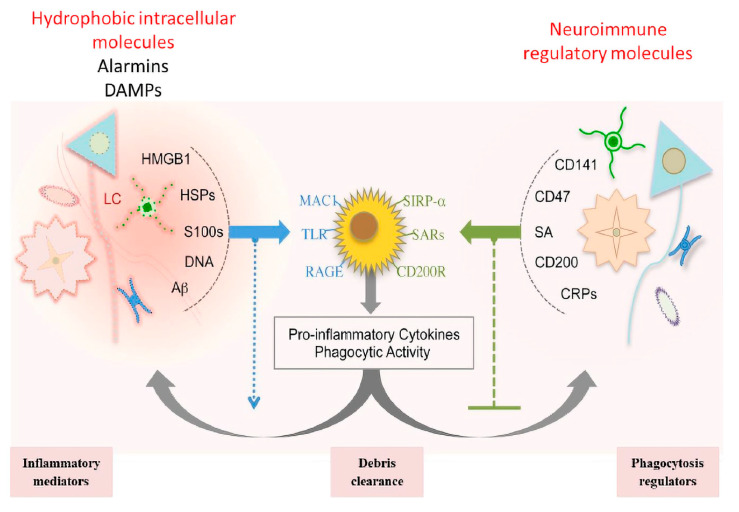
Molecular interactions balancing inflammatory responses, debris clearance and phagocytosis regulators following SCI with left showing hydrophobic intercellular molecular interactions controlling harmful signal while right cycle reflect neuro-inflammatory molecular interaction controlling phagocytosis while center portion show multimolecular interactions to clear cellular phagocytic debris.

**Figure 6 ijms-21-07533-f006:**
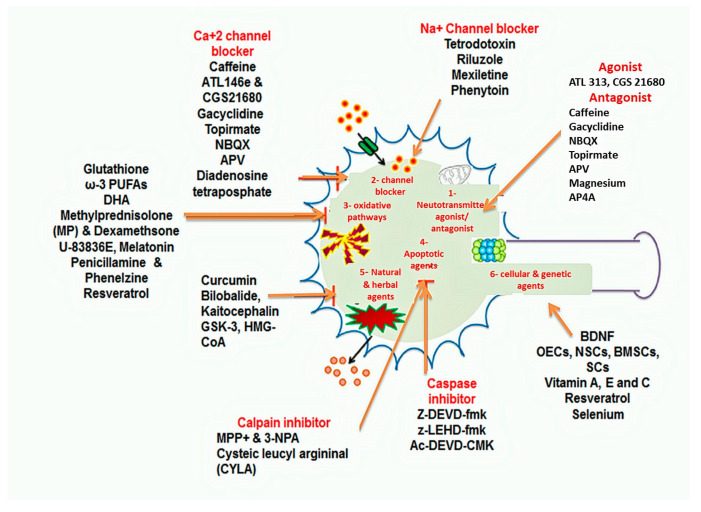
Neuroprotective pathways and different neuroprotective approaches with centre portion showing neuroprotective pathways (i) neurotransmitter agonist/antagonist, (ii) channel blockers, (iii) anti-oxidative pathways, (iv) apoptotic pathway (v) herbal and natural agents, (vi) cellular and genetic agents, while various agents acting on specific pathways are shown by pointed arrows.

**Figure 7 ijms-21-07533-f007:**
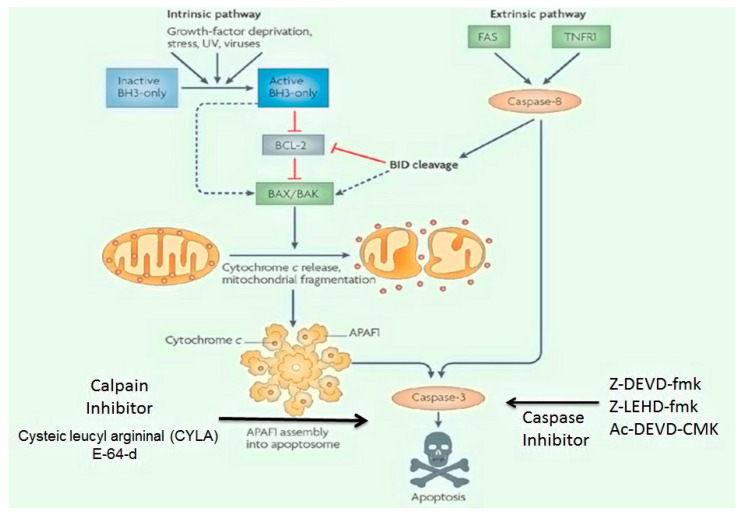
Apoptotic pathway (i) intrinsic pathway and (ii) extrinsic pathway with anti-apoptotic inhibitors, i.e., calpain and caspase that act on specific target molecule and retard apoptosis.

**Figure 8 ijms-21-07533-f008:**
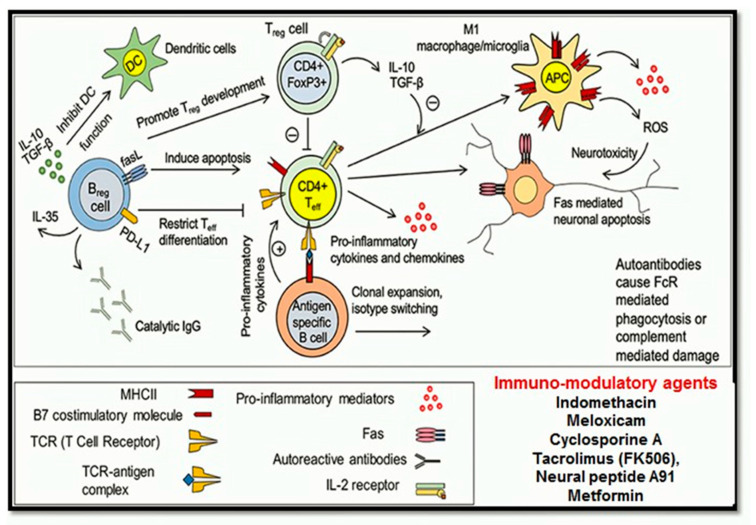
Immuno-modulatory (neuro-inflammatory) pathway following spinal cord injury and specific immuno-modulatory agents.

**Figure 9 ijms-21-07533-f009:**
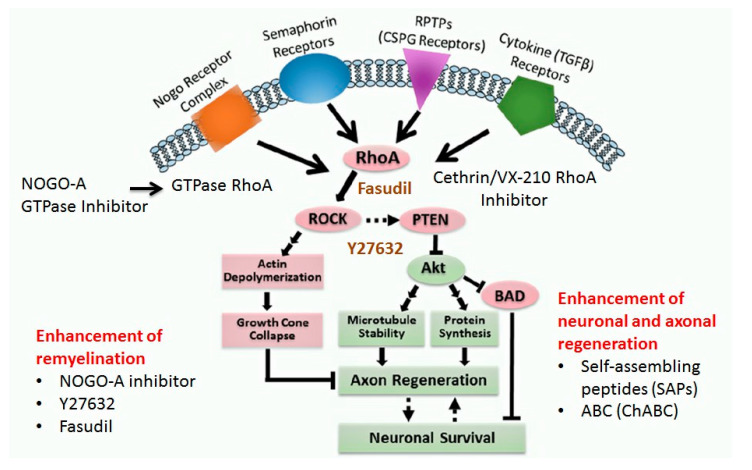
Neuroregenerative pathway (RhoA/Rho and Rock pathway) and underlying neuroregenerative approaches (i) enhancement of remyelination and (ii) enhancement of neuronal and axonal regeneration strategies.

**Table 1 ijms-21-07533-t001:** Intercellular Signaling Molecules involved in cellular responses: phases, source, class and regulatory functions in SCI.

Phases	Signaling Molecules	Source	Class	Function	Ref.
Phase I	Thrombin	Serum	Protease	Clot formation &astrocyte proliferation	[50]
	ATP	Neuron, oligodendrocytes & astrocytes	Neurotransmitters	Microglia chemotaxis & reactive astrogliosis	[41,42,43]
	Glutamate	Neuron, oligodendrocytes & astrocytes	Neurotransmitters	Microglia chemotaxis & reactive astrogliosis	[52]
Phase I & II	Alarmins (HMGB1)	Damaged cells	DAMPs	Pro-inflammatory signals & increase phagocytosis	[53]
	S100s				[53]
	DNA				[53]
	PAMPs (LPS)	Microbes			[54]
	IL-1b, TNFa, INFg	Astrocyte, microglia & Leukocytes	Cytokines Chemokines	Pro-inflammatory regulation	[53]
	IL-6, CCL2			Leukocyte instruction & astrocyte scar formation	[52]
	CD200, CD47	Neurons	NIRegs	Protection of healthy self	[53]
Phase III	Neurotrophins & BDNF	Neurons & Astrocytes	Neural Remodeling	Synapse remodeling	[54]
	Thmbs, C1q	Astrocytes & Microglia		Synapse formation & pruning	[55]
	Perineuronal net	Astrocytes & O progenitor cell		Restrict terminal sprouting	[55]
Phase I & III	MMP-9	Serum & Microglia		OPC proliferation, remyelination & neovascular remodeling	[55]
	Kallikreins	Astrocytes, Microglia, Neurons & Serum	Proteases	Proinflammatory & demyelination	[56]
	Serpins	Astrocytes, Microglia & O progenitor cells		Inhibit deleterious protease	[55]
	FGF	Astrocytes, Neuron & Endothelia	Growth Factors & Morphogens	Fibrotic scar, ECM & neovascular remodeling	[41,43]
	VEGF	Endothelia, Fibroblast & Astrocytes		Neovascular remodeling & remyelination	[55]
	PDGF-B	Endothelia & Astrocytes			
	PDGF-A				
Phase II & III	Endothelin, EGF, BMP	Neuron, Astrocytes & O progenitor cells	Growth Factors, Morphogens	Astrocyte proliferation & glial scar formation	[57,58]

**Table 2 ijms-21-07533-t002:** Commonly use glutamate receptor agonist/antagonist; NMDA, AMPA & kainate receptor antagonist as a neuroprotective approaches.

Sr. No.	Compound	Class	Receptor	Mechanism of Action	Reference
1	Gacyclidine (GK-11)	Tenocyclidine, closely related to phencyclidine	Noncompetitive NMDA receptor	Inhibits formation of ischemic SCI lesion.	[62]
2	NBQX	2, 3-Dihydroxy-6-nitro-7-sulfamoylbenzoquinoxaline	AMPA/kainate receptor antagonist	Enhances mitochondrial functions and retard ROS and RNS formation.	[63]
3	Topirmate	2,3:4,5-Bis-O-(1-methylethylidene)-beta-D-fructopyranose sulfamate	AMPA receptor antagonist	Promotes neuroprotective activity, improves tissue recovery, oligodendrocytes and motor function. NBQX and topiramate both showed powerful neuroprotective activity in female rat model.	[63,64]
4	APV	2- Amino-5-phosphovaleric acid	NMDA receptor antagonist	Block glutamate activation and transport.	[64]
5	Magnesium	element	Non-competitive NMDA receptor antagonist	Reduces excitotoxicity and inflammation.	[9]
6	AP4A (Diadenosine tetraposphate)	Putative alarmone	Pirinergic receptor partial agonists and even act as antagonists in presence of the full agonist of P2 receptors (P2 are ATP receptors)	Reduces ATP-dependent excitotoxicity related death by both lowering the intracellular calcium response and decreasing the expression of P2 receptors.	[65]

**Table 3 ijms-21-07533-t003:** Channel blockers i.e., Na^+^ and Ca^2+^ channel blockers that have a potential neuroprotective activity.

Sr. No	Compound	Class	Group	Mechanism of Action	Ref.
1.	Tetrodotoxin (TTX)	Guanidine	Na^+^ channel blocker	TTX block cellular Na^+^/Ca^+2^ exchange, membrane depolarization, and glutamate release and block neuronal cell death. TTX also improve motor function.	[66]
2.	Riluzole	Benzothiazole	Voltage-gated Na^+^ channel blocker	Inhibit glutamate transmission and reduces glutamate associated excitotoxicity in neuronal tissue. Stop Na^+^ efflux and H+ influx within neurons and prevent neuronal acidosis.	[68,69,70]
3.	Mexiletine	local anesthetic, antiarrhythmic agent, similar to lidocaine	Na^+^ channel blocker	Stop demyelination of neuronal tissues and evoke motor function recovery. Decreases lipid peroxidation, evokes motor function and promote neuroprotection.	[70]
4.	Phenytoin	Hydantoin derivative	Na^+^ channel blocker	Block cellular Na^+^/Ca^+2^ exchange and promote neuroprotection	[70]
5.	Nimodipine	Dihydropridinic	L-type VGCCs	Reduces malondialdehyde (MDA) levels, macrophages marker ED1activation and activation of myeloperoxidases (MPo). Prevent oxidative damage by reduction of macrophages infiltration to injured tissues.	[71]
6.	Mibefradil	Posicor	T-type VGCCs	Selective blockade of transient, low-voltage-activated (T-type) calcium channels	[72]
7.	Trimethadione	oxazolidinedione	T-type VGCCs	Selective blockade of transient, low-voltage-activated (T-type) calcium channels	[72]

**Table 4 ijms-21-07533-t004:** Herbal and natural extracts with neuroprotective activity.

Sr. No.	Compound	Class	MOA	Ref.
1.	Bilobalide	Terpenoids from Ginkogo biloba leaves extract	Showed neuroprotective action on neurons and schwann cells by inhibiting ROS formation and apoptosis, It also modifies cytochrome-C oxidase subunit I level and regulates mitochondrial functions	[100]
2.	Centella asiatica (L.) Urban (CA)	*(pegaga)* malay & Chinese traditional medicine	It acts as a brain tonic, which improve memory, it was also found to improve spinal cord recovery in organotypic rat model	[101]
3.	MLC601 & MLC901	*Neuroaid*	It is a combination of natural products, that has shown to be safe and to aid neurological recovery after brain & spinal injuries and have a potential role in improving recovery after SCI	[103]
4.	Kaitocephalin	*Eupenicillium shearii* extract	Potent glutamate receptors (AMPA & NMDA) antagonist and inhibit glutamate excitotoxicity	[104]
5.	Myricetin	Flavonoid	Inhibits glutamate excitotoxicity by stopping NMDAR receptor phosphorylation and reducing Ca^+2^ overloads	[105]
6.	Curcumin	Curcuminoids of turmeric (Curcuma longa)	Exert neuroprotective activity by restoration of glutathione S transferase (GST), glutathione peroxidases (GPx) and MnSOD (manganese superoxide dismutase) activity	[102]

**Table 5 ijms-21-07533-t005:** Immunosuppressive or immunomodulatory drugs commonly reported to use during SCI.

Sr. No.	Compound	Class	MOA	Ref.
1.	Indomethacin	Non-steroidal anti-inflammatory drug (NSAID) is a nonselective cyclooxygenase inhibitor (COX)	It inhibits prostaglandin production and prevents tissue necrosis. Indomethacin prevents RhoA synthesis (RhoA prevents axonal growth), prevent oligodendrocytes loss and axonal myelination.	[140,142]
2.	Meloxicam	COX2 inhibitor	It inhibits prostaglandin synthesis, reduces oxidative stress and provides neuroprotection by inhibiting the production of ROS, LPO, GSH and DNA fragmentation.	[143]
3.	Cyclosporine A	Immunosuppressant	It inhibits helper T lymphocytes, cytotoxic and inflammatory responses in macrophages, expression of nitric oxide synthase, production of tumor necrosis factor (TNF-α) and reduce expression of IL-1, IL-2, and IL-6	[139]
4.	Tacrolimus (FK506)	Immunosuppressant (isolated from *Streptomyces tsukubanensis*)	It possesses neuroprotective effect on T cells and modulates inflammation. It also inhibits caspase-3, NF-kB and promotes oligodendroglial survival.	[140]
5.	A91 (87-99 immunogenic sequence)	Neural peptide INDP	It promotes neuroprotection by activating T-lymphocytes, Th2 anti-inflammatory activity and promote brain-derived neurotropic factor (BDNF). INDP inhibits iNOS expression, ON production and LPO generation after SCI prevents apoptosis.	[141,142]
6.	Metformin	Hypoglycemic drug, AMP-protein kinase (AMPK), an agonist.	It inhibits apoptosis by inhibiting mTOR and p70S6K pathways, promote autophagy and inhibit NF-kB inflammation. It also regulate TNFα and IL-1β inflammatory cytokines	[142]

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
