# Peer review of "Spinal Cord Injury: Pathophysiology, Multimolecular Interactions, and Underlying Recovery Mechanisms"

_ijms, 2020, doi:10.3390/ijms21207533_

Round 1

Reviewer 1 Report

The manuscript is an extensive review of SCI.

It is well structured, with a huge information of pathophysiology, cellular and molecular interactions. It is a well-organised review from the topic.

I have some comments to improve or make the text more understandable:

Line 187: if you agree, probably I suggest to add “assisting in inflammation, repair and recovery processes” or similar.

Line 291: “endogenous mesenchymal cells enter…” probably you have to add a citation for that.

Line 322-329: In my opinion, probably you can add a sentence where explain this balanced inflammatory response in terms of: a first stage of proinflammatory response and a later anti-inflammatory response, to have more comprehensible. The idea that inflammatory response is necessary is in the text (since too extreme response such as slower response in inflammation results in cytotoxins accumulation and higher inflammatory response results in cellular damage), but probably can be re- rewritten in this way of proinflammatory and anti-inflammatory balance. It is just a suggestion.

Line 491 and 498: From all molecules described in the text, should be impossible to have a table with which treatments are used in clinical, because should be more pages and work, and the review is quite complete right now, but at least you can comment the updated guidelines issued in 2013 by the CNS and the American Association of Neurological Surgeons (AANS), that recommend against the use of steroids early after an acute SCI. In line 498 is about safety and efficacy of MP, but you can include some reference about recommendation from AANS, or describe the discrepancies in the literature and clinical.

Lines 730 and 743: The idea from lines 322-329 are well explained here. I have a suggestion to include a small sentence about IL-4 and a reference. Talking about IL-10, I suggest to include “delayed administration of IL-4 (48h after SCI) markedly improves functional outcomes and reduces tissue damage after contusion injury”.

Francos-Quijorna, I., Amo-Aparicio, J., Martinez-Muriana, A., and López-Vales, R. (2016). IL-4 drives microglia and macrophages toward a phenotype conducive for tissue repair and functional recovery after spinal cord injury. Glia 64, 2079–2092. doi:10.1002/glia.23041.

In my opinion is a significant fact to take in consideration to be included.

Figure 4: In my opinion is a figure not really easy to understand. Probably I’m wrong but I don’t know if could be expressed in other way.

Figure 6: This one needs a small revision for sure. Into the cell we can find the pathways from 2 to 6, but #1 is missing into the cell, at least the title, like “1. Neurotransmitter agonist”, and the comas “,” I think are in wrong place in the figure text, just to revise.

Overall recommendation is to accept for publication after these minor suggestions.

Author Response

Dear Reviewers, thank you for your constructive comments and input. Here I have attached the point-to point response to your queries (in bold). Line numbers are also indicated where applicable. New changes in the manuscript are highlighted in yellow. Hope it will meet your expectations and we manage to answer your queries well. 

********

  • Line 187: if you agree, probably I suggest adding “assisting in inflammation, repair and recovery processes” or similar.

Thank-you for your comment yes, I agree with your comment and I have added the word inflammation in the text. Line 194 and it is highlighted.

  • Line 291: “endogenous mesenchymal cells enter…” probably you have to add a citation for that.

Dear Reviewer I really appreciate your concern, I have added the reference for the mentioned statement in line # 304, reference # 42.

  • Line 322-329: In my opinion, probably you can add a sentence where explain this balanced inflammatory response in terms of: a first stage of proinflammatory response and a later anti-inflammatory response, to have more comprehensible. The idea that inflammatory response is necessary is in the text (since too extreme response such as slower response in inflammation results in cytotoxins accumulation and higher inflammatory response results in cellular damage), but probably can be re- rewritten in this way of proinflammatory and anti-inflammatory balance. It is just a suggestion.

The point you suggested is really important and I have now explained it briefly as balanced between pro & post inflammatory responses. I have explained in term of M1 & M2 inflammatory responses. Moreover, I have re-written it to make it more clear, line 334-341. More detail about M1 and M2 responses is discuss under neuroinflammation line 810-815.

  • Line 491 and 498: From all molecules described in the text, should be impossible to have a table with which treatments are used in clinical, because should be more pages and work, and the review is quite complete right now, but at least you can comment the updated guidelines issued in 2013 by the CNS and the American Association of Neurological Surgeons (AANS), that recommend against the use of steroids early after an acute SCI.

Thank you for this suggestion, yes I have added more text to explain the recommendation for use of corticosteroid after 8 h of acute injury (after 24-48 h) and also explain about assigned limitations regarding the clinical administration, according to the latest AANS & CNS guidelines, published in 2013 and also other guideline published in 2017, line 527-535.

  • In line 498 is about safety and efficacy of MP, but you can include some reference about recommendation from AANS, or describe the discrepancies in the literature and clinical.

Brief explanation and new reference is added about the safety and efficacy of MP administration according to AANS & CNS guidelines, line 540-546. 

  • Lines 730 and 743: The idea from lines 322-329 are well explained here. I have a suggestion to include a small sentence about IL-4 and a reference. Talking about IL-10, I suggest including “delayed administration of IL-4 (48h after SCI) markedly improves functional outcomes and reduces tissue damage after contusion injury”.

“Francos-Quijorna, I., Amo-Aparicio, J., Martinez-Muriana, A., and López-Vales, R. (2016). IL-4 drives microglia and macrophages toward a phenotype conducive for tissue repair and functional recovery after spinal cord injury. Glia 64, 2079–2092. doi:10.1002/glia.23041”

Yes, I have included some sentences in my text explaining more about IL-4 activity and its role in neuroinflammation, using the stated reference, line 815-817.  

  • Figure 4: In my opinion is a figure not really easy to understand. Probably I’m wrong but I don’t know if could be expressed in other way.

Yes, probably you are right; now I have tried to make it more understandable, as, this figure show the three lesion compartments and multicellular interactions associated with it. So, I have changed it accordingly. Hope it works now.

  • Figure 6: This one needs a small revision for sure. Into the cell we can find the pathways from 2 to 6, but #1 is missing into the cell, at least the title, like “1. Neurotransmitter agonist”, and the comas “,” I think are in wrong place in the figure text, just to revise.

In this figure I use a general term as receptor agonist and antagonist, but yes it is better to add the same word as I have stated in text. So, I have revised the figure 6 and its caption as per suggestion.

These are our best responses to your queries and comments. Thank you very much for your time in reviewing our paper and providing valuable suggestions to improve the manuscript.

Yours sincerely,

YOGESWARAN LOKANATHAN

Reviewer 2 Report

I suggest strong improvement of the work!!!!

This manuscript does not contain any novel findings. It misses some important aspects relative to therapeutic approches in development such as bioengeneering or/and cell therapies...I would like also to highligh that the most recent papers striclty focused on SCI are dated 2012!!! there are some papers of 2019 but too much general!

Moreover, the iconography (fugures) require more attention...the style is very old!Figures seem to be cut and paste from a thesis.

As a last comment ther is not a methodological paragraph explaining the methods applied for revision of the literature!!!

Author Response

Dear Reviewers, thank you for your constructive comments and input. Here I have attached the point-to-point response to your queries (in bold). Line numbers are also indicated where applicable. New changes in the manuscript are highlighted blue. We hope that you will get satisfied with our explanation, related to your concerns and queries. 

**********

This manuscript does not contain any novel findings. It misses some important aspects relative to therapeutic approaches in development such as bioengineering or/and cell therapies.

Thank you for your comments. I have briefly explained cell therapies under cellular and genetic approaches, as stem cell therapies. But I agreed to you and I have added more detail about cellular therapies (line 735-737 & 747-751) and also bioengineering therapies for SCI is also briefly described in the text, line 752-767.

I would like also to highlight that the most recent papers strictly focused on SCI are dated 2012!!! There are some papers of 2019 but too much general!

Thank-you for your comment just to clear this, most of the references I added in my review article fall within the time frame of 2009-2020. However, now some references I have replaced with latest one with the similar content i.e. 11, 23, 27 & 31.  

Moreover, the iconography (figures) require more attention...the style is very old! Figures seem to be cut and paste from a thesis.

Thank you for this comment I have improved the Figures and tried to make it more presentable.

As a last comment there is not a methodological paragraph explaining the methods applied for revision of the literature!!!

Thank you for your concern. However, I just want to highlight that this is a narrative review not the systematic review and as the common practice the narrative review do not require any methodological paragraph and explanation.

These are our best responses to your queries and comments. Thank you very much for your time in reviewing our paper and providing valuable suggestions to improve the manuscript.

Yours sincerely,

YOGESWARAN LOKANATHAN

Round 2

Reviewer 2 Report

The authors have improved the manuscript as requested.